# How Can Systematic Reviews Teach Us More about the Implementation of the 3Rs and Animal Welfare?

**DOI:** 10.3390/ani9121163

**Published:** 2019-12-17

**Authors:** Merel Ritskes-Hoitinga, Judith van Luijk

**Affiliations:** SYstematic Review Center for Laboratory (Animal) Experimentation, Department for Health Evidence, Radboud University Medical Center, 6525 GA Nijmegen, The Netherlands; Judith.vanLuijk@radboudumc.nl

**Keywords:** three Rs, systematic reviews, research quality, reporting, evidence-based

## Abstract

**Simple Summary:**

The three Rs stand for Replacement, Reduction, and Refinement of animal studies and were published for the first time in 1959 by Russell and Burch. Replacement refers to avoiding the use of (non-human) animals in research. Reduction implies using fewer animals, for example, by better statistical methods and better literature studies, and Refinement means reducing the discomfort and improving the welfare of animals used in experiments. The three Rs have gained more interest and popularity since the 1970s, and have now become the crucial central element in the revised legislation in Europe, the EU Directive 2010/63EU, controlling the proper use of animals in experiments in the European member states. Animals are used in order to improve the health and welfare of other non-human animals, in veterinary medicine, and of humans, for toxicological purposes and in clinical medicine. Using animals in experiments has always been subject to ethical and societal debate. At Syrcle, we have introduced the methodology of systematic reviews for preclinical animal studies since 2012. This methodology comes from the clinical field and is a key element in evidence-based medicine, as systematic reviews summarise the scientific evidence as objectively as possible. A systematic review (SR) is defined as a literature review focused on a single question that tries to identify, appraise, select, and synthesise all high-quality research evidence relevant to that question. Introducing this methodology for the preclinical animal studies seems very logical, as animal studies in clinical medicine are performed for protecting humans from ineffective or unsafe treatments. Systematic reviews thus lead to summarising evidence from preclinical studies before entering clinical trials. In addition to protecting humans, systematic reviews can also be used to implement the three Rs. Examples of how systematic reviews contribute to implementing the three Rs are provided in the following article, thus demonstrating the value for protecting animals as well.

**Abstract:**

This paper describes the introduction of the systematic review methodology in animal-based research and the added value of this methodology in relation to the 3Rs and beyond. The 3Rs refer to Replacement, Reduction, and Refinement of animal studies. A systematic review (SR) is defined as a literature review focused on a single question that tries to identify, appraise, select, and synthesise all high-quality research evidence relevant to that question. Examples are given on how SRs lead to the implementation of the 3Rs and better science. Additionally, a broader context is given regarding societal, political, and scientific developments. Various examples of systematic reviews are given to illustrate the current situation regarding reporting, quality, and translatability of animal-based research. Furthermore, initiatives that have emerged to move further towards more responsible and sustainable research is of benefit for both animals and humans.

## 1. Introduction

The 3Rs [1] have been enormously important towards developing good quality science, safeguarding animal welfare and aiming to reach reproducible and reliable scientific results. When the word “animal” is used in this paper, it refers to non-human animals. In the field of Replacement, it has been shown that through the use of human cardiomyocytes, the prediction of arrhythmic effects of drugs became more accurate [2]. Reducing the number of animals is achieved by providing food restriction instead of ad libitum feeding, as it leads to reduced variation in experimental results, as well as increased health and life span [3]. A longer life span also contributes to a reduction in animal numbers needed at the start of long-term toxicity testing [4]. The timing of providing the restricted amount of food is essential, as this needs to be adapted to the species specific needs and circadian rhythms (Refinement), in order to prevent stereotypic behaviour and adverse outcomes in physiological parameters and responses to pharmacological agents [3]. While virtually everybody will agree on the importance of the 3Rs, the implementation has turned out to be quite a challenge and is considered not effective enough [5]. Questionnaires sent out to researchers and animal welfare officers on the implementation of the 3Rs indicated that the importance of the 3Rs is acknowledged, but at the same time, it is indicated that already existing 3Rs possibilities have probably not been found and thus not implemented [6,7]. Legislation in Europe requires that existing 3R possibilities must be implemented (EU Directive 2010/63EU), so strictly speaking, not implementing already existing 3R possibilities would be illegal.

One of the challenges we have encountered is the search for the 3Rs, as there is a huge number of 3R databases and websites available, each with different content, structure, and search strategy. This makes the 3R search almost like a ‘mission impossible’. Except for the 3R search, the finding of an alternative does not automatically imply that it will be implemented, due to the general challenge of making a change. Moreover, there is a clear difference between the 2Rs Refinement and Reduction and the R of Replacement. Replacement often requires a long-term complex strategy, which also involves regulators in the process, whereas Reduction and Refinement can often be implemented immediately in the short-term in individual experiments. So, there is not a one size fits all solution, and in addition, the process of Replacement is also very complex and demanding, encompassing many stakeholders, high costs, political will and perseverance.

In 2006, a 3R Research Center was founded within the Central Animal Laboratory at Radboud University Medical Center, with the aim to provide support to researchers to search for and identify the 3Rs and aim to implement them in the daily practice of the animal unit [8]. In the first two years, a subsidy from the license holder was received, and during that period, 60 questions related to the 3Rs were dealt with. The results were variable, and depending on the particular topic and situation, Reduction and/or Refinement could be implemented in individual studies. In general, customers indicated they were satisfied with this service. After two years, the subsidy from the license holder stopped, and researchers would then have to pay a fee for the same 3R service. From then on, zero questions were received. By interviewing researchers, the conclusion was that there was no money available in the projects to pay for a 3R service. This provided food for thought, as the 3Rs are a legal requirement when doing animal studies, but this was obviously not included in the budgets of the projects. This also indicates that the 3Rs are not considered a high priority when no money is made available for this topic. It was decided to send out questionnaires to learn more about the process of implementing the 3Rs in practice [6,7]. On the basis of the results from the questionnaires and a national workshop on 3R implementation [9], it was decided that the methodology of systematic reviews would have to be developed and adopted for the preclinical field, as this would not only lead to the 3R implementation but also stimulate better quality science and translational transparency [5].

## 2. Political and Societal Developments

In 2010, the revised EU Directive 2010/63EU was installed, where the 3Rs play a central role. As a result of the Citizen’s initiative against animal testing and a public hearing by the European Commission, the conclusion was that more education on the 3Rs is key for its implementation. In Europe, it has recently been acknowledged that the educational curricula do not provide sufficient/effective 3R teaching and training, as the implementation of the 3Rs in practice is not considered effective enough. This has been expressed by a number of tenders on pilot projects initiated by the European Parliament and Commission and coordinated by the EURL Joint Research Center, and DG Environment. These pilot projects are currently ongoing and, for example, focus on a more effective 3R implementation in educational curricula at the high school, university, and continuous education level, and will provide e-learning on a number of 3R topics, as well as create a platform (ETPLAS: Education and Training Platform in Laboratory Animal Science) where all the 3R training materials will be made available.

In the Netherlands, the parliament and government have set a goal for the transition towards animal-free innovations, thus especially promoting Replacement. In this respect, as a first step, the Netherlands aim for phasing out animal testing for research on the safety of chemicals by 2025. Many stakeholders are involved in discussions on how this goal can be achieved, and in November 2019, an international conference was organised, as one country cannot achieve this goal alone. Also, within the EU, phasing out animal studies has been set as a goal, but a date for achieving this goal has not yet been determined.

So, the political arena in Europe seems to quickly move towards Replacement and animal-free innovations. The motivation is based partly on avoiding the use of animals, and thus animal suffering in research, but maybe even more importantly, that the alternative models using human materials can translate better to humans. Systematic reviews are considered animal-free innovations as they lead to new results and insights without doing new animal studies. Except for the advantages for the laboratory animals, they also lead to translational transparency of the value of animal studies for human and veterinary medicine.

## 3. Systematic Reviews Methodology

A systematic review (SR) is defined as a literature review focused on a single question, which tries to identify, appraise, select, and synthesise all high-quality research evidence relevant to that question. An SR comprises several steps: 1. Phrase the research question, 2. Search for all evidence, 3. Select the relevant studies, 4. Extract the study characteristics (species, sex, age, etc.), 5. Assess the study quality and/or do a risk of bias assessment, 6. Do a (statistical) analysis of all results found (meta-analysis). The SR methodology is well-known from evidence-based medicine and has been performed routinely for clinical trials since 1992 when the Cochrane Collaboration was founded. The mission of the Cochrane Collaboration (www.cochrane.org) is to promote evidence-informed health decision-making by producing high-quality, relevant, accessible SRs and other synthesised research evidence. There are currently about 11,000 members and more than 68,000 supporters from more than 130 countries worldwide. The SR methodology leads to a complete overview of already available evidence, as objectively as possible. Many steps are done by two independent reviewers, and discrepancies solved by discussions and/or a third reviewer. This method differs from narrative reviews because the methodology guarantees that literature is analysed as completely and objectively as possible. With narrative reviews, it can easily happen that a selective choice of publications is made supporting opinions, which can lead to the wrong scientific conclusions and hypotheses, leading to research waste that could have been prevented (see, for example, Steven A Greenberg [10], with selective referencing). In an SR, a comprehensive search needs to be conducted, so scientific proof of all the evidence is identified and analysed. It is also quite logical to do preclinical SRs, as these studies are claimed to be done with the aim to obtain information on safety and efficacy, and thus protect humans. When planning new animal studies, it is also useful to conduct SRs, and in the next section on the 3R implementation, concrete examples will be discussed. The SR methodology leads to transparency on the quality of publications and on the translational value of preclinical studies for human and veterinary medicine.

SYRCLE (SYstematic Review Center for Laboratory (animal) Experimentation) was founded in 2012 and has been dedicated to developing teaching and research in the field of preclinical SRs (www.syrcle.nl). In order to facilitate and accelerate the process, tools and guidelines have been developed. A protocol has been developed, which needs to be filled out (and preferably published) before the start of the SR [11]. By completing a protocol in advance of conducting an SR, this seeks to reduce bias during the conduct of an SR, as it can be tempting to adapt the procedure on the basis of new information found. For doing effective literature searches, search filters for animal studies in Pubmed and Embase have been developed [12,13,14,15]. By using these filters and combining these with the particular topic of interest, all animal studies published in that area and presented in Pubmed and Embase will be found. SYRCLE has also developed a Risk of bias tool [16], and has published guidance on how to perform a meta-analysis [17].

## 4. Benefits of Systematic Reviews for Subsequent Human Clinical Trials

By doing SRs, a transparent overview of all available evidence is created, thereby incorporating the reliability of the information found by judging the quality and/or risk of bias. This overview also leads to the identification of knowledge gaps and translational transparency. For example, by not adapting to the basic science principles of randomisation and blinding, Sena and Van der Worp [18] have demonstrated that an overestimation of the effect of a drug can be found. For stroke and Multiple Sclerosis (MS), SRs have indicated a huge number of positive results from animal studies, but only a few treatments have been successful in the clinic [19,20]. As far as we know, the first SR on animal studies was performed by a Dutch MD, Janneke Horn [21]. In a clinical trial in Stroke patients, examining the effect of the drug Nimodipine, neutral results were found. The trial had been based on positive results of this drug in animal studies. After the trial, it was then decided to do a retrospective SR of all animal studies, and this also showed no positive effect of this drug anymore. This suggests that SRs of preclinical studies can help prevent exposing human patients to ineffective drugs unnecessarily. A similar finding was done for adverse effects in clinical trials: a clinical trial resulted in serious adverse effects, which had not been seen in individual animal studies, but a retrospective SR showed the same effects in animals [22]. Another recent example on the use of a tuberculosis (TB) booster vaccination with a neutral result in a clinical trial retrospectively revealed the same result in the SR of the animal studies [23]. An intensive investigation by the British Medical Journal also suggested that animal studies had been used selectively for obtaining funding and ethical approval. This raises a very important general question on how animal studies are used and evaluated in medical practice. If animal studies are selectively performed and/or used, this is not a scientifically objective nor valid method, and cannot be in accordance with the 3Rs either. In a paper by Wieschowsky [24], an analysis of investigator brochures demonstrated that the information provided did not make a rigorous evaluation possible of the preceding animal studies. If animal studies are really done with the purpose of protecting humans, then that process should be taken more seriously. SRs of preclinical animal studies seem to be the first step to take.

Except for the sake of protecting humans, SRs have also shown to be an important aid in implementing the 3Rs and improving animal welfare.

## 5. Examples of Implementation of the 3Rs and the Impact for Animal Welfare

When doing SRs, the extensive search can make unnecessary duplication transparent. An SR on intestinal anastomosis research found that 88 out of a total of 1342 studies had duplicated the adverse effect of chemotherapy on wound healing [25]. This adverse effect of chemotherapy on wound healing is a well-known effect, and future new animal studies on this topic can, therefore, be avoided. The study by Yauw [25] also highlighted that in 83% of the publications it was not mentioned whether surgical technique had been performed under sterile conditions, and 91% of the publications did not mention whether postoperative analgesia had been applied. In the cases where these elements were not mentioned, it is not clear whether these had been applied or not. Within Cochrane, it is the standpoint that when it has not been mentioned, it has not been done. If that would be true for these animal studies, there is serious reason for concern on how animals in experiments are treated and, thus, serious concerns for animal welfare and interference with experimental results. From the point of view of Refinement, it is of the utmost importance that these procedures on sterile surgery and postoperative analgesia are implemented in the correct manner and reported.

Cumulative meta-analyses are meta-analyses that are updated when new studies on a certain topic are published. In a cumulative meta-analysis by Sena on the effect of the drug tPA in stroke [26], after a certain date, new animal studies did not lead to a change of the effect of the drug. This implies that new animal studies on the effect of the drug after a certain date could have been prevented.

An SR by Currie et al. [27] focused on the use of animals for studying bone cancer pain. Mechanically evoked pain behaviours were most commonly reported; however, the largest differences in results occurred when studying spontaneous pain behaviours. Spontaneous pain behaviours are considered important outcome measures for clinical relevance and for the development of effective therapeutic targets. From the point of view of animal welfare, measuring spontaneous behaviours is associated with lower degrees of discomfort as compared to evoked pain behaviour. This illustrates that SRs can also be useful for the ethical evaluation of planned animal studies as it aids in assessing the severity of procedures in relation to the relevance of the outcome parameters for the goal of the study.

In the case of making a selection of an (animal) model system, systematic reviews support more evidence-based choices. For tissue engineering experiments on cartilage defects, de Vries et al. [28] showed that rodents and rabbits are frequently used and that the pathophysiological processes in these models, however, do not reliably resemble the human process. Cartilage defects in humans are often limited to the cartilage, making healing problematic as there is no blood vessel ingrowth. Due to the anatomical features, experimental cartilage defects in rodents and rabbits are always osteochondral, which means extending into the bone underneath the cartilage, leading to blood vessel ingrowth, and successful healing. So, using rodents and rabbits will almost ‘automatically’ lead to positive results of any treatment. However, as this is a different process as compared to humans, larger animal species are, therefore, more suitable. Recently, the use of veterinary patients as models for human medicine has gained popularity, the so-called one health/one medicine concept. By studying veterinary patients having similar cartilage defects as in humans, such as horses, also relevant therapeutic results for humans can be obtained. From the point of view of animal welfare, as well as scientific reasons, the one health/one medicine concept is a very promising road to take.

Zeeff et al. [29] reviewed animal models in the field of inflammatory bowel disease and discovered that the models that resemble the human process the most were hardly used. This suggests that many studies without a huge translational relevance have been executed and published. With the choice of animal models in science, we regularly see that this is based on high profile publications and/or tradition, and making a change in the model system has major implications for a research institute. However, if the model is used for a certain human disease, then the choice of model ought to be based on evidence that this is the most optimal model for that particular disease. A conflict between Refinement and Reduction can occur because the choice for the most optimal model for inflammatory bowel disease can imply that animal welfare is compromised more, as this model resembles the chronic form of the disease. However, the advantage is that many animal studies with little to no relevance can be saved at the same time.

SRs can also be done on other studies than animal studies, such as in vitro studies [30]. By assembling the evidence from these studies that are often based on the use of human materials, it can be expected that animal studies can be prevented. The use of SRs for alternatives to animal testing holds great promise for the future, as it can provide the evidence for the validation of alternative tests.

Within basic science, SRs are regularly considered of no use, as they would not lead to new results and ideas. In an SR in basic neuroscience, new results and insights were obtained, which may also prevent new animal studies on that particular topic [31].

When doing SRs, in general, fewer animal studies are being done, because the time is spent otherwise. When introducing preclinical SRs at Radboud university medical centre, a 35% reduction in animal use was obtained for the period 2006–2014, while maintaining a similar output (number of publications). As a comparison, a 15% reduction in animal use in the Netherlands occurred during that same period. In 2017, SYRCLE was awarded the second Cochrane reward prize because of the contribution to ensuring value and reducing waste in research: www.cochrane.org/news/cochrane-reward-prizes-reducing-waste-2017-winners.

The unfortunate benefit of preclinical systematic reviews is making transparent that the quality of reporting of preclinical (animal) studies is largely inadequate, demonstrating that there is huge room for improvement.

## 6. Quality of Reporting

All SRs of animal studies demonstrate a lack of reporting of essential details. Fifty to eighty per cent of animal studies do not mention randomisation and blinding. Randomisation is one of the basic starting points of our scientific practice and is a method to assign study subjects to an experimental group based on chance alone, thus making sure individuals are equally distributed over the groups. Blinding is another scientific principle to reduce potential bias, which implies that a researcher does not know which treatment is given to which individual subjects. The ARRIVE (Animal Research Reporting of In Vivo Experiments) guidelines for reporting were published in 2010 and have been endorsed by over a 1000 scientific journals; however, the actual implementation remains a challenge [32]. Because the ARRIVE guidelines have been developed for the publication process, which is at the latest stage in the research process, it can be too late to make the necessary changes. The PREPARE (Planning Research and Experimental Procedures and Animals: Recommendations for Excellence) guidelines have been developed, as these can be used during the planning stages, which is a more optimal timing [33]. In 2017, it was examined whether filling out a checklist on ARRIVE during the publication process would help to improve publication quality (IICARus study). Filling out the checklist hardly led to any improvements in mentioning details in the publications [34]. It is, therefore, of the utmost importance that journal editors, reviewers, funders, and researchers start working together for the actual improvement of planning, executing, and reporting quality for the sake of science and (non-human) animal and human welfare [35].

Recent initiatives that hold promise for positive change are the formulation of guiding principles for health research by the Ensuring Value in Research Funder Forum (EVIR) (www.ensuringvalueinresearch.org). EVIR has formulated 10 guiding principles for funding clinical trials, with the aim to implement those in the funder’s practice, leading to more value in research and a reduction of waste. Those guiding principles have been redrafted for the preclinical field as well with SYRCLE in the lead [36], and will be further developed, discussed, and implemented among the members of the funder forum and other stakeholders. As an example, principle 2 states: “Research should only be funded if set in the context of one or more existing systematic reviews of what is already known or an otherwise robust demonstration of a research gap. Currently, new research is often not justified (not robust) on what is known, e.g., unnecessary duplication.” Another principle reads: “All studies should report methods and findings in full (and Open Access), following credible and justifiable reporting guidelines (e.g., ARRIVE). This applies irrespective of the nature of the findings (e.g., unexpected negative/neutral results), the way they are reported and whether the study was completed as planned.” When these guiding principles for funding preclinical studies are going to be implemented, this will ensure more value in research, of benefit for both (non-human) animals and humans.

A recent European consortium, EQIPD (European Quality in Preclinical Data), encompassing 29 partners from academia and industry, is working towards developing a general and flexible management quality system to be applicable in animal units in both industry and academia. A more general quality standard for animal studies will help to set the scene for improving preclinical research quality that will indeed be applied in practice.

## 7. Food for Thought

In the past 60 years, the 3Rs have had a major influence on animal use in research. New insights and emerging challenges, such as ‘the reproducibility and translational crisis’ and ‘research waste’, show that these principles have not lost their relevance. However, we need innovative approaches, such as the systematic review methodology to create transparency on quality and translation, in order to continue a successful transition towards more responsible and sustainable research, thereby implementing the 3Rs and achieving much more.

## 8. Conclusions

There is already abundant evidence that preclinical systematic reviews make a major contribution to improving quality of science and translational transparancy, as well as to implementing the 3Rs Replacement, Reduction and Refinement. In the development towards non-animal alternatives, systematic reviews can also play a decisive role in the discovery, development and validation of these alternatives. In order to ensure more value in research, it appears imperative to require the conduct of preclinical systematic review for the preparation of new animal and human studies.

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
