# Peer review of "How Can Systematic Reviews Teach Us More about the Implementation of the 3Rs and Animal Welfare?"

_animals, 2019, doi:10.3390/ani9121163_

Round 1

Reviewer 1 Report

In my view I think the authors make a compelling case that systematic review of pre clinical trial animal studies is important. The major focus is of course on animal welfare and the 3Rs. However, I also thought the authors made a compelling case that systematic review of the animal studies could provide important insights for later human clinical trials. Perhaps it would be appropriate to include a separate section with its own sub heading where the authors detail the benefits for systematic review of animal studies for subsequent human clinical trials. This material already exists in the manuscript, but I think the case could be made even more compelling if this material were placed in a section on its own.

I do have an issue with much of the language. I am fairly confident that English is not the first language of authors. I suggest (with some humility as I am sure they can write better in my language than I can in theirs!) that it would be enormously helpful to find an editor to work on the language. I started to make specific editorial improvements but gave up because of the large number of sentences that needed work. 

Specific feedback on manuscript

Line 14 - the legislation referred to is almost 10 years old, it is not new?

Line 17 - animals are not just used in clinical research but in testing for carcinogens for example

Line 24 "Except for" - should this be in "in addition to"?

Lines 38-39 The 3Rs are clearly important for animal welfare but I think the claim that they are important for good science needs to be supported by argument. I am not saying the view is wrong, it just needs to be explained. 

Lines 43-44 Not sure what is meant by "already existing 3 Rs"

Line 58 It is stated that help need to "find the 3Rs" again now sure what is meant here

Line 127 - it is stated that the methodology prevents bias. This is a very strong statement. The methodology seeks to reduce bias but there are numerous biases in what gets published in the frist place. 

Author Response

We wish to thank reviewer 1 for the positive feedback that a compelling case is made.

We have inserted a new subheading on the benefits of systematic reviews for clinical trials, as suggested.

The editor will will update the English language.

Line 14. "new" has been changed in revised.

Line 17. we have added toxicological purposes.

Line 24. Changed into "in addition to"

Lines 38-39. Explanations are provided on how the 3Rs also lead to good science.

Lines 43-44. Sentence has been modified.

Line 58. Text has been modified.

Line 127. Text has been reformulated.

Reviewer 2 Report

The authors make a case for using the methodology of systematic reviews in preclinical animal studies to promote implementation of the three Rs (Replacement, Reduction, and Refinement). I found their examples in sections 3 and 4 compelling. My specific comments are detailed below.

Although the English is generally okay, there are still some places that require checking by someone whose first language is English. For example, the phrase “learn us more” in the title of the paper should be changed (line 1) and lines 32-34 require editing. Ideally, the entire paper should be checked.

Lines 16-17: Throughout the manuscript “humans” are differentiated from “animals”, but because humans are animals, the distinction should be between “humans” and “non-human animals”. See also lines 243 and 259.

Abstract: I suggest adding information to the Abstract to provide readers with more context and a better idea of the paper’s content. A definition of “systematic review” would be good to include here.

Line 48: What is meant by “build up”?

Line 65 requires editing – see “B interviewing”.

Line 109: Readers who are not from Europe may not be familiar with the Cochrane Collaboration (or Cochrane), so provide a brief description.

Throughout the paper, the reference number in brackets should directly follow the name(s) of the author(s). For example, Line 141, Janneke Horn is mentioned, but the reference number, [18], is not found until line 144. Line 154, Wieschowsky is mentioned, but the reference number, [21], is found at the very end of the sentence rather than right after the author’s name. Line 166: insert [22] right after Yauw, so it is clear which study you are describing. Lines 176-177: move [23] to right after Sena. Lines 190-192: move [25] to right after de Vries et al. and correct the spelling of “pathophysiological”. Lines 203-204: move [26] to right after Zeeff et al.

Line 224: Does “nr” mean “number”? If so, spell out number.

Lines 227 and 245: spacing issues.

Lines 229-230: This sentence is difficult to understand, especially the phrase “unfortunate benefit”. Please clarify.

Lines 232-233: Write out numbers at the start of sentences and delete “e.g.”. I suggest that you briefly define randomization and blinding here. Write out in full the first time “ARRIVE” is used. Do the same for PREPARE (line 237).

Line 261: “Consortium” is singular, so the verb should be “is” not “are”.

References: Please check Instructions for Authors on reference formatting (e.g., article titles should not be italicized but abbreviated titles for journals should be italicized and include periods, such as Lab Anim.).

Author Response

We wish to thank reviewer 2 for the valuable comments. 

The English language will be revised by the editor. 

Lines 16-17. In the introduction it is indicated that when the wording "animals" is used, this refers to non-human animals. 

Abstract has been extended to give a better idea of the content and context. Also the definition of systematic review has been added. 

Line 48. "Build up" has been changed in "structure".

Line 65. B interviewing means By interviewing.

Line 109. A description of the Cochrane has been provided. 

Reference numbers have been added following the names of the authors.

Line 224. Number has been spelled out. 

Lines 229-230. Sentence has been reformulated.

Lines 232-233. Modifications done.

Line 261. Verb "is" inserted. 

References have been checken, however not sure if correct output style has been used now.